# Ion fluxes Involved in the Adaptation of the Estuarine Diatom *Coscinodiscus centralis* Ehrenberg to Salinity Stress

**DOI:** 10.3390/ijms241813683

**Published:** 2023-09-05

**Authors:** Changping Chen, Xiao Hu, Yahui Gao, Junrong Liang, Lin Sun

**Affiliations:** 1Key Laboratory of Ministry of Education for Coastal and Wetland Ecosystems, School of Life Sciences, Xiamen University, Xiamen 361102, China; xhu2023@126.com (X.H.); sunljr@xmu.edu.cn (J.L.); 2State Key Laboratory of Marine Environmental Science, Xiamen University, Xiamen 361102, China; sunlin@xmu.edu.cn

**Keywords:** estuarine diatom, salinity, ion flux, the scanning ion-selective electrode technique

## Abstract

Although estuarine diatoms have a wide range of salt tolerance, they are often severely stressed by elevated salt concentrations. It remains poorly understood how estuarine diatoms maintain ionic homeostasis under high-salinity conditions. Using a scanning ion-selective electrode technique, this study determined the fluxes of H^+^, Na^+^, and K^+^ involved in the acclimatization of the estuarine diatom *Coscinodiscus centralis* Ehrenberg after an elevation in salinity from 15 psu to 35 psu. The *C. centralis* cells exhibited marked H^+^ effluxes after a transient treatment (TT, 30 min) and short-term treatment (ST, 24 h). However, a drastic shift of H^+^ efflux toward an influx was induced in the long-term treatment (LT, 10 days). The Na^+^ flux under TT, ST, and LT salinity conditions was found to accelerate the Na^+^ efflux. More pronounced effects were observed under the ST and LT salinity conditions compared to the TT salinity condition. The K^+^ influx showed a significant increase under the LT salinity condition. However, the salinity-induced Na^+^/H^+^ exchange in the estuarine diatom was inhibited by amiloride and sodium orthovanadate. These results indicate that the Na^+^ extrusion in salt-stressed cells is mainly the result of an active Na^+^/H^+^ antiport across the plasma membrane. The pattern of ion fluxes under the TT and ST salinity conditions were different from those under the LT salinity conditions, suggesting an incomplete regulation of the acclimation process in the estuarine diatom under short-term salinity stress.

## 1. Introduction

Diatoms are able to live in waters containing different concentrations of dissolved salts, from freshwater to brackish and marine waters [1]. Individual diatom taxa have characteristic salinity optima and ranges, and particular groups of diatoms seem to prefer environments with specific levels of salinity [2,3]. Freshwater diatom species are negatively affected by an increase in salinity [4,5], whereas oceanic species fail to grow under low-salinity or freshwater conditions [6]. However, estuarine diatom species often demonstrate a broad tolerance to salinity and correspond well with the salinities observed in their natural habitats [7]. Although they seem to be euryhaline, the reproduction rates in many estuarine taxa are inhibited by an increase in salinity [6,8]. Salinity is often considered an important determinant in the distribution of diatoms in estuaries [9]. The succession of diatoms along the estuarine salinity gradient has generally been ascribed to the fact that these species suffer salinity stress upon exposure to enhanced salt concentrations [10,11]. 

Salinity has an ion toxic effect on cells because the high intracellular concentrations of chloride and Na^+^ are deleterious to cellular systems [12,13]. Moreover, cellular ion homeostasis can be disturbed by a permanent influx of inorganic ions [14]. Accordingly, maintaining a low concentration of salt in the cytosol is of the utmost importance in cells’ tolerance to salinity stress. The internal Na^+^ concentration in the green alga *Dunaliella salina* (Dunal) Teodoresco does not change significantly during salt acclimation, indicating that Na^+^ ions were actively exported out of the cytoplasmic space against the electrochemical Na^+^ gradient by cells living under elevated salt concentrations [15]. The active export of Na^+^ to the apoplast or external environment is essential for sustaining Na^+^ homeostasis in the cytosol. This is chiefly carried out by Na^+^/H^+^ antiporters secondarily energized by the proton motive force, which is generally generated by a plasma membrane (PM) H^+^-ATPase [16]. The activity of PM Na^+^/H^+^ antiporters has been reported in microalgae and cyanobacteria [17,18]. Therefore, H^+^ pumping is fundamental to the Na^+^/H^+^ exchange and salinity stress. However, the active Na^+^/H^+^ antiport across the PM and its contribution to salt exclusion in microalgae are still lacking. Aside from its function in salt acclimation, the Na^+^/H^+^ antiporter plays a principal role in intercellular pH regulation [19] and becomes active when intracellular acidification is induced [20].

K^+^ homeostasis also plays a crucial role in the salt adaptation of microalgae [21,22]. Levels of intracellular K^+^ increased significantly with increases in salinity in the diatom *Chaetoceros muelleri* Lemmermann [23]. The intracellular K^+^ content in *D. salina* cells is kept fairly constant over a wide range of salinities, suggested that the cells possess efficient mechanisms to eliminate Na^+^ and accumulate K^+^ and that the intracellular concentrations of Na^+^ and K^+^ are carefully regulated [24]. The reduction in the intracellular K^+^ pool is often correlated with a massive K^+^ efflux and a significant impairment of cell metabolism, and this K^+^ efflux is initiated within seconds of acute NaCl stress and may last for several hours [25,26].

Estuarine diatoms are subjected to salinity variations influenced by periodical tides, and temporary salt increases may change their growth conditions in an unfavorable manner, such as through the accumulation of toxic ions [2]. The diatoms’ acclimation to the altered internal ion situation is managed by the transport of ions at the plasma membrane [27]. Despite the importance of ion homeostasis, there is only a limited understanding of the fluxes at the plasma membranes of microalgae under saline conditions. *Coscinodiscus centralis* Ehrenberg is a dominant species in the Jiulong River Estuary, and it is a good material for studying how estuarine diatoms maintain ionic homeostasis under high-salinity conditions. In this study, we measure the fluxes of H^+^, Na^+^, and K^+^ in the estuarine diatom *Coscinodiscus centralis* Ehrenberg under saline conditions, using a non-invasive ion flux technique. The aim of this study was to compare the alternations in ion fluxes in an estuarine diatom with different exposure times to an enhanced salt concentration.

## 2. Results

### 2.1. Salinity Optima for Growth in Cultures

The effects of different salinities on the growth response curves of the estuarine diatom *C. centralis* are shown in Figure 1. *C. centralis* were able to reproduce between salinities of 10 and 35, and the optimum salinity was 15 psu. *C. centralis* responded similarly to the 10 psu, 15 psu, and 20 psu treatments before day 6. After that, differences among the growth responses in three treatments were pronounced. However, the growth of *C. centralis* was more strongly inhibited by 35 psu than the other treatments from the beginning.

### 2.2. Response of F_v_/F_m_ to Salinity Stress

The influences of TT salinity, ST salinity, and LT salinity on *F_v_/F_m_* are presented in Figure 2. Compared to the control treatment, the TT-stressed, ST-stressed, and LT-stressed cells showed significantly lower *F_v_/F_m_* values. However, no significant changes were observed among the TT-stressed, ST-stressed, and LT-stressed cells. It is interesting that the *F_v_/F_m_* value showed a small increase after 24 h of salinity stress. We also measured the valve diameters of *C. centralis* in three treatments. The diameters of *C. centralis* ranged from 52.9 to 71.0 μm, from 54.9 to 74.9 μm, and from 56.2 to 74.5 μm, with mean diameters of 62.2 ± 5.0 μm, 65.0 ± 5.7 μm, and 64.6 ± 5.5 μm for the TT, ST, and LT treatments, respectively.

### 2.3. Ion Fluxes in Response to Salt Stress in the Estuarine Diatom C. centralis: H^+^ Fluxes

For the control treatment, the scanning ion-selective electrode technique (SIET) data showed a stable and constant efflux, with a mean value of 5.7 × 10^−3^ pmol cm^−2^ s^−1^ (Figure 3A). There were marked differences in H^+^ fluxes with salinity between the TT-stressed, ST-stressed, and LT-stressed cells. The TT-stressed cells exhibited a drastic H^+^ efflux, ranging from 13.7 × 10^−3^ to 17.4 × 10^−3^ pmol cm^−2^ s^−1^ (Figure 3B). In the ST-stressed cells, the H^+^ efflux decreased rapidly after 24 h from 6.3 × 10^−3^ to 9.4 × 10^−3^ pmol cm^−2^ s^−1^ (Figure 3C), while a net H^+^ influx was observed in the LT-stressed cells that ranged from −4.1 × 10^−3^ to −1.3 × 10^−3^ pmol cm^−2^ s^−1^ (Figure 3D). In general, the H^+^ flux displayed significant responses to the salinity stresses in the four treatments (Figure 3E).

### 2.4. Na^+^ Fluxes 

As shown in Figure 4, there was a balance between the Na^+^ efflux and influx in the control treatment (Figure 4A). A typically greater Na^+^ efflux was observed in the TT-stressed cells compared to the control over the period of recording, although it did not respond significantly in both treatments (Figure 4B). However, salinity induced a marked Na^+^ efflux in the ST-stressed cells, and the effect was even more pronounced in the LT-stressed cells (Figure 4C,D). Following the cells’ exposure to the medium containing 35 psu for 24 h and 10 days, the mean Na^+^ efflux rate in the ST-stressed and LT-stressed cells increased 16-fold and 34-fold, respectively (Figure 4E).

### 2.5. K^+^ Fluxes

A K^+^ influx with a mean value of −36.6 pmol cm^−2^ s^−1^ was recorded in the control treatment (Figure 5A). Following a slight decrease in the TT-stressed cells, the K^+^ influx recovered to its normal value in the ST-stressed cells after 24 h (Figure 5B,C). However, compared to the control cells, an accelerated K^+^ influx was observed in the LT-stressed cells in which the mean flux rate increased to −61.6 pmol cm^−2^ s^−1^(Figure 5D,E).

### 2.6. Effects of PM Transport Inhibitors on H^+^, Na^+^, and K^+^ Fluxes

Since a clear Na^+^/H^+^ exchange was exhibited in the treatment with 35 psu for 10 days, LT-stressed cells were used to characterize the effects of PM transport inhibitors on the cell H^+^, Na^+^, and K^+^ fluxes. Amiloride (25 μM), the specific inhibitor of the Na^+^/H^+^ antiporter, or sodium orthovanadate (100 μM), the specific inhibitor of PM H^+^-ATPase, significantly inhibited the Na^+^ efflux and K^+^ influx in the LT-stressed cells (Figure 6 and Figure 7). The PM transport inhibitors amiloride or sodium orthovanadate significantly reduced the salinity-induced H^+^ efflux in the TT-stressed treatment (data not shown), although no significant effect was observed in the LT-stressed cells (Figure 8). However, cells exhibited an unstable H^+^ flux with the addition of inhibitors. Particularly, a pronounced shift toward an H^+^ efflux was seen after cells were subjected to amiloride (Figure 8B).

## 3. Discussion

### 3.1. Growth under Salinity Stress

Our results showed that the estuarine diatom *C. centralis* demonstrated optimum growth at 15 psu (Figure 1). Many studies show that optimal salinity levels in cultivated diatom strains correspond well to the salinities of the samples from which the strains were isolated [4,7,28]. For example, oceanic strains have a salinity optimum at 33 psu, compared to estuarine species at 15 psu [4]. Diatoms from thalassic hypersaline environments demonstrate optimal or near-optimal growth rates at salinities as high as three times that of seawater [28]. Brand (1984) also reported that many oceanic species die at 45 psu, while all of the estuarine species are able to reproduce at 45 psu, indicating their differences in salinity acclimation [4]. In the open oceans, salinity varies between 33 psu and 37 psu, while in estuaries, the salinity may range from 0 psu to the full strength of seawater. Some field studies show that acclimating to changes in salinity is a prerequisite for most diatoms living in estuaries and coastal wetlands [7,9]. 

Although some studies show that estuarine diatoms grow well over a broad range of salinity [4,8,29], the growth of the estuarine diatom *C. centralis* at a salinity of 35 psu was significantly less than at 15 psu in our study, indicating that the diatom experienced salinity stress with an increase in salinity (Figure 1). Interestingly, the translocation of chloroplasts to the center of the cell was observed in the LT-stressed cells. The movement of chloroplasts has been reported in diatoms and is induced by high light irradiance, contact stimulation, and electric stimulation [30,31,32]. It is suggested that chloroplast migration may help to maintain photosynthetic activity or to protect the nucleus under unfavorable conditions [31]. Our results showed that the *F_v_/F_m_* ratio was significantly inhibited in the TT, ST, and LT-stressed cells (Figure 2). Therefore, estuarine diatoms may suffer from frequent short-term increases in salinity. Using a SIET in this study, we concluded that ion fluxes in TT and ST salinity differed from ion fluxes in LT salinity, indicating an incomplete regulation of the acclimation process in the estuarine diatom under short-term salinity stress.

### 3.2. Na^+^/H^+^ Antiport across the PM under Salinity Stress

In this study, a net H^+^ efflux was observed in the control (Figure 3). This is in agreement with the results that marine phytoplankton, such as diatoms, coccolithophores, and dinoflagellates, often show an outward net proton motive force at their plasma membranes [19,33]. The H^+^ flux at the plasma membrane plays an important role in the regulation of the intracellular pH. Taylor et al. (2011) showed that coccolithophores possess a voltage-gated H^+^ channel which removes H^+^ rapidly from the cell during calcification and helps maintain a constant intracellular pH [19].

Although protons play many roles in signaling, development, and metabolic regulation, physiological and genomic evidence supports a Na^+^-energized plasma membrane in marine phytoplankton [33]. The high extracellular Na^+^ concentration in the open ocean environment can be utilized by marine phytoplankton to drive coupled transport processes across the plasma membrane. This differs from embryophytes and filamentous fungi, which generally utilize electrogenic PM H^+^-ATPase to energize secondary transport at the plasma membrane [34]. However, our results suggested that PM H^+^-ATPase pumps protons and maintains electrochemical H^+^ gradients, thus promoting a secondary active Na^+^/H^+^ antiport at the PM during the process of the estuarine diatom *C. centralis* acclimating to salinity stress. The experimental evidence is briefly listed below. TT and ST salinity both caused a net H^+^ efflux in *C. centralis*, although the more pronounced effect was observed in the TT-stressed cells (Figure 3C,D). A drastic shift in the H^+^ efflux toward an influx was induced after 10 days of salinity stress. A salinity-induced H^+^ influx was also seen in plant cells [16]. H^+^-SO_4_^2−^ symporters have been identified in the diatom *Phaeodactylum tricornutum* Bohlin, which may inhabit brackish waters, although there are several reports of the Na^+^-coupled uptake of nutrients such as nitrate, ammonia, and silicon in marine diatoms [35,36]. On the other hand, under the conditions of TT, ST, and LT salinity, the Na^+^ flux was found to accelerate the Na^+^ efflux (Figure 4). More pronounced effects were observed under the conditions of ST and LT salinity compared to TT salinity (Figure 4E). Moreover, pharmacological experiments suggested the involvement of PM H^+^-ATPase in Na^+^/H^+^ antiport. The PM transport inhibitors amiloride (an inhibitor of the Na^+^/H^+^ antiport) and sodium orthovanadate (an inhibitor of PM H^+^-ATPase) decreased the H^+^ efflux in TT-stressed cells (data not shown). The rectification of the Na^+^ flux was correspondingly reversed when the inhibitors affected the H^+^ influx in LT-stressed cells (Figure 6 and Figure 8). In this study, the changes in the H^+^ flux corresponding to the Na^+^ efflux suggest that Na^+^ extrusion in salt-stressed cells is mainly the result of an active Na^+^/H^+^ antiport across the PM. 

Salinity induced a marked Na^+^ efflux in ST- and LT-stressed cells (Figure 4), indicating the maintenance of a low internal Na^+^ concentration in *C. centralis*. A similar low Na^+^ concentration has been reported in *Dunaliella* under high-salinity conditions [24]. Although it has been clearly established that glycerol plays a crucial role in cell osmoregulation, ion homeostasis would help to regulate the ionic composition of *Dunaliella* cells over a wide range of salinities [37,38]. The almost unchanged NaCl concentrations in *Dunaliella* clearly indicate that organisms living under elevated salt concentrations actively export Na^+^ ions out of the cytoplasmic space against the electrochemical Na^+^ gradient [15]. The characterization of the Na^+^/H^+^ antiporter from the plasma membrane of *Dunaliella* suggests that this is chiefly carried out by Na^+^/H^+^ antiporters secondarily energized by the proton motive force. The Na^+^/H^+^ antiporter has been characterized in all types of organisms including microalgae [18,20]. Given these results, we suggest that Na^+^ extrusion in salt-stressed *C. centralis* cells is correlated with an increase in the activity of PM Na^+^/H^+^ antiporters.

### 3.3. K+ Fluxes under Salinity Stress

A slight decrease in K^+^ influx was observed in the cells after a treatment with 35 psu for 30 min, indicating an increase the K^+^ efflux (Figure 7). This is in agreement with previous study, which found that early under high-salinity conditions, *Dunaliella* cells export K^+^ into the surrounding medium [39]. In plant cells, one of the cellular responses to salt stress is a massive K^+^ efflux [25]. Such a K^+^ efflux is initiated within seconds of acute NaCl stress and may last for several hours, reducing the intracellular K^+^ pool and significantly impairing cell metabolism [26]. Maintaining a cytosolic K^+^ concentration in an environment with a high Na^+^ concentration is a key factor in determining the ability to tolerate salinity [40]. In this study, the K^+^ influx recovered in ST-stressed cells, indicating that the intracellular K^+^ concentration in *C. centralis* would not necessarily change during this period. However, the K^+^ influx showed pronounced increase after treatment with 35 psu for 10 days (Figure 7). The dominance of K^+^ against Na^+^ in the cytoplasm and the fast uptake of K^+^ after an upshock are fundamental characteristics of bacterial salt acclimation [41]. Pick et al. (1986) found that limiting the external K^+^ concentration causes an increase in intracellular Na^+^ and a decrease in intracellular K^+^ in *Dunaliella salina* under high-salinity conditions [24]. A transport mechanism exchanging K ions for Na ions has been already suggested in *Platymonas* and *Chlorella pyrenoidosa* [2,37]. Therefore, we conclude that increasing the K^+^ influx helped in the regulation of the ionic composition of *C. centralis* cells under LT salinity.

## 4. Conclusions

The alternations in the ion fluxes in the estuarine diatom with different exposure times to enhanced levels of salinity were compared in present paper. We found that *C. centralis* cells exhibited marked H^+^ effluxes after TT and ST treatment. However, a drastic shift of the H^+^ efflux toward an influx was induced in the LT treatment. Under the TT, ST, and LT salinity conditions, the Na^+^ flux was found to accelerate the Na^+^ efflux. The K^+^ influx showed a significant increase under the LT salinity condition. Our results indicate that the Na^+^ extrusion in salt-stressed cells is mainly the result of an active Na^+^/H^+^ antiport across the plasma membrane. The patterns of ion fluxes in the TT and ST salinity conditions were different from those under LT salinity, suggesting an incomplete regulation of the acclimation process in the estuarine diatom under short-term salinity stress.

## 5. Materials and Methods

### 5.1. Culture Conditions

The *C. centralis* strain (MMDL50816) used in this study was isolated from the estuary of the Jiulong River in Fujian Province, China, in 2013 (Figure 9) [42]. The *C. centralis* cells were maintained in artificial seawater with a salinity of 15 psu [43]. The cultivation temperature was 20 °C, the light/dark cycle was 12:12 h, and the light intensity was 60 μmol photons m^−2^ s^−1^.

### 5.2. Determination of Optimum Salinity 

To investigate its optimum salinity, the estuarine diatom *C. centralis* was tested at 10 psu, 15 psu, 20 psu, and 35 psu after isolation. These salinities were chosen because they reflect the conditions under which it was sampled. Artificial sea water [43] at different salinities was prepared for the diatom culture. the salinity was measured via a refractometer (S-100 portable salinometer, YAMATO, Kanagawa, Japan). The cells were pre-adapted to different salinities before the start of the experiment. Growth experiments were carried out in duplicate cultures for all the salinities measured. The cell number was determined for each sample in a Coulter counter with an inverted light microscope at ×200–400 magnification. 

### 5.3. Treatments

Cells cultured in artificial seawater with a salinity of 15 psu were used as the control treatment. Then, the cells were subjected to a transient treatment (TT) with 35 psu seawater for 30 min, a short-term treatment (ST) with 35 psu seawater for 24 h, and a long-term treatment (LT) with 35 psu seawater for 10 days. NaCl stock (1M) was slowly added to the medium (15 psu) until the salinity reached 35 psu. With the increase in salinity stress, a range of chloroplast translocation was observed from relatively little chloroplast translocation (ST salinity) to the movement of the majority of chloroplasts to the center of the cell (LT salinity).

### 5.4. Measurement of Maximal Quantum Yield of Photosystem II (F_v_/F_m_) under Different Salinity Stresses

The *F_v_/F_m_* value was measured via a Xe-PAM fluorometer (Walz, Effeltrich, Germany). For fluorescence measurements, the parameters of the fluorometer were set to ensure that samples from each culture were dark-adapted for 30 min. The minimal fluorescence (F0) was measured at a low light intensity, and additional background laser-light was used for the measurement of maximal fluorescence (Fm).

### 5.5. Measurements of Net H^+^, Na^+^, and K^+^ Fluxes via a Scanning Ion-Selective Electrode Technique (SIET)

Prior to the ion flux measurement, the cells were first fixed on the bottom of the measuring chamber to reduce the mobility caused by the movement of the electrode during the measurements. Briefly, glass slips containing 0.1% poly-l-lysine (Yue Xu Sci. and Tech. Co. Ltd.) were placed in a measuring chamber containing a measuring solution of H^+^, Na^+^, and K^+^. Then, 50 L cell suspensions were placed in the middle of the poly-l-lysine pretreated coverslips. Most cells had been settled on the surface after 10 min. The coverslips were then washed with the measuring solution in order to remove the unsettled cells. The ion fluxes were measured after 3 ml of a measuring solution of H^+^, Na^+^, and K^+^ was slowly added to the measuring chamber. The steady-level ion flux measurements were initiated and continued for 7 to 8 min. H^+^, Na^+^, and K^+^ were measured in the following solutions, respectively: (1) H^+^: 8 mM KCl, 5 mM CaCl_2_, 20 mM MgCl_2_, 180 mM NaCl, 10 mM Na_2_SO_4_, 10mM HEPES, pH 8.2; (2) Na^+^ and K^+^: 0.1 mM KCl, 0.1 mM CaCl_2_, 0.1 mM MgCl_2_, 0.5 mM NaCl, 10 mM HEPES, pH 8.2.

After their exposure to the saline (TT, ST, and LT), subsamples were collected to obtain ion flux measurements. To decrease the influence of the release of salt on the ion flux recordings (preloaded Na^+^ would diffuse from the surfaces of salinity-stressed cells in a buffer containing lower Na^+^ concentrations), the cells were placed in the measuring solution to equilibrate for 15 min. Then, the flux rate decreased gradually and reached a steady state within 10 min. The measuring solutions were removed slowly from the measuring chamber with a pipette, and 3 ml of a fresh solution was slowly added.

We investigated the inhibitory effects of PM transport inhibitors on the ion fluxes in *C. centralis*. LT-stressed cells were subjected to 100 μM of sodium orthovanadate (a PM H^+^-ATPase inhibitor) or 25 μM of amiloride (a Na^+^/H^+^ aniporter inhibitor) for 15 min in the measuring solution. Then, measuring solutions containing sodium orthovanadate were replaced with 3 mL of fresh measuring solution before measurement. Since the amiloride had no obvious effect on the Nernstian slopes of the H^+^, Na^+^ and K^+^ electrodes, measuring solutions containing amiloride were not replaced.

The net fluxes of H^+^, Na^+^, and K^+^ were measured noninvasively using an SIET system (NMT-100, Younger USA LLC, Amherst, MA 01002, USA) [16,44]. The H^+^, Na^+^, and K^+^ microelectrodes were prepared as follows: pre-pulled and silanized glass micropipettes were first filled with a backfilling solution (H^+^: 40 mM KH_2_PO_4_ and 15 mM NaCl, pH 7.0; Na^+^: 250 mM NaCl; K^+^: 100 mM KCl) to a length of approximately 1.0 cm from the tip, and the fronts of the micropipettes were filled with 15 µm columns of selective liquid ion-exchange cocktails (H^+^ LIX: Fluka 95293, XY-SJ-H, Younger USA; Na^+^ LIX: Fluka 71178, XY-SJ-H, Younger USA; K^+^ LIX: Fluka 60398, XY-SJ-H, Younger USA). An Ag/AgCl wire-electrode holder (EHB-1, World Precision Instrument, Sarasota, FL, USA) was inserted into the back of the electrode to make an electrical connection with the electrolyte solution. A DRIREF-2 (World Precision Instrument, Sarasota, FL, USA) was used as the reference electrode. Ion-selective electrodes were calibrated prior to the flux measurements as follows: H^+^—pH 7.8, 8.2, and 8.7; Na^+^—0.05 mM, 0.5 mM, and 5 mM; and K^+^—0.01 mM, 0.1 mM, and 1.0 mM. The concentration gradients of the target ions were recorded by moving an ion-selective microelectrode between two positions close to the cell. Image and data recording, preliminary processing, and the movement of electrodes controlled by a stepper motor were carried out using the SIET system. Electrodes with a response of more than 50 mV per decade Nernstian slopes for H^+^, Na^+^, and K^+^ were used in our study. The ion flux was calculated via Fick’s law of diffusion, as follows:J = −D (dc/dx)(1)
where J is the ion flux in the x direction, dc/dx represents the ion concentration gradient, and D represents the ion diffusion constant in a particular medium. 

### 5.6. Data Analysis

Three-dimensional ionic fluxes were calculated using JCal V3.2 software, which was developed by Yue Xu Sci. and Tech. Co., Ltd. (Beijing, China). The positive values in the figures represent ion efflux and vice versa. The difference of individual treatments within an experiment was assessed using a one-way ANOVA and Tukey’s HSD post-hoc test. For all treatments, the homogeneity of variances was verified, and significance was determined at *p* < 0.05. All statistical analyses were performed using SPSS version 16.0 (SPSS Inc., Chicago, IL, USA). 

## Figures and Tables

**Figure 1 ijms-24-13683-f001:**
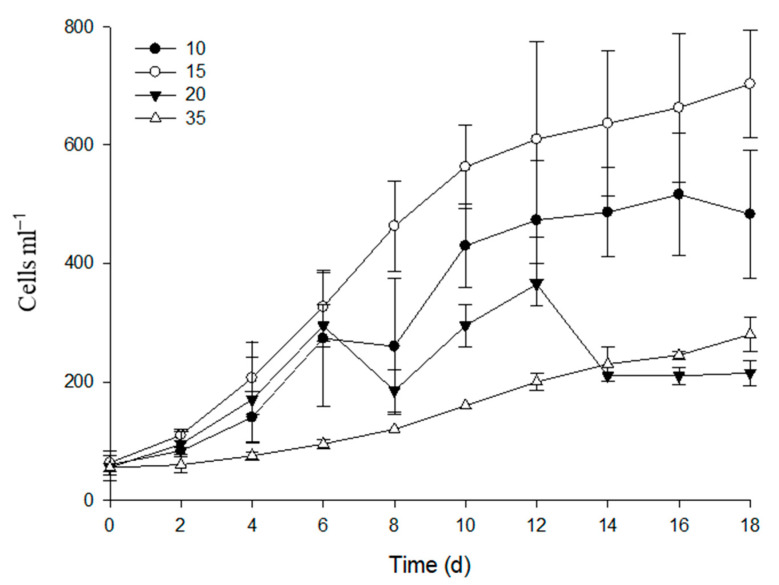
Effects of different levels of salinity on the growth of the estuarine diatom *Coscinodiscus centralis*.

**Figure 2 ijms-24-13683-f002:**
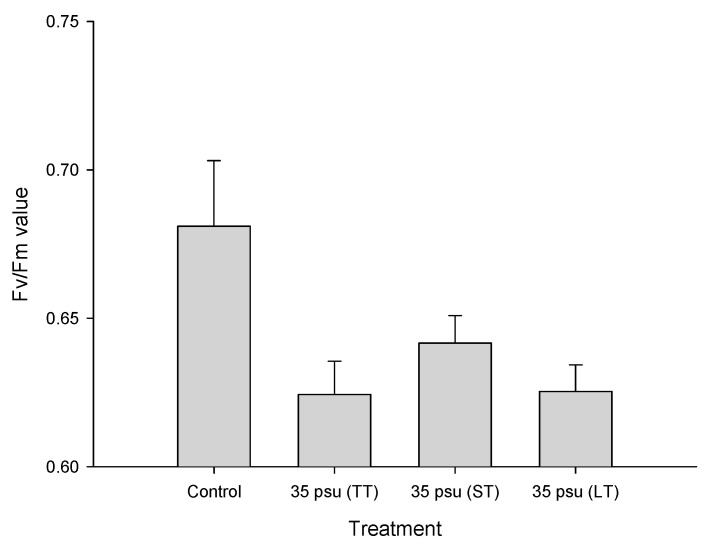
Effects of transient-treatment (TT) salinity (35 psu for 15 min), short-term (ST) salinity (35 psu for 24 h) and long-term (LT) salinity (35 psu for 10 days) on the *F_v_/F_m_* value of the estuarine diatom *Coscinodiscus centralis*.

**Figure 3 ijms-24-13683-f003:**
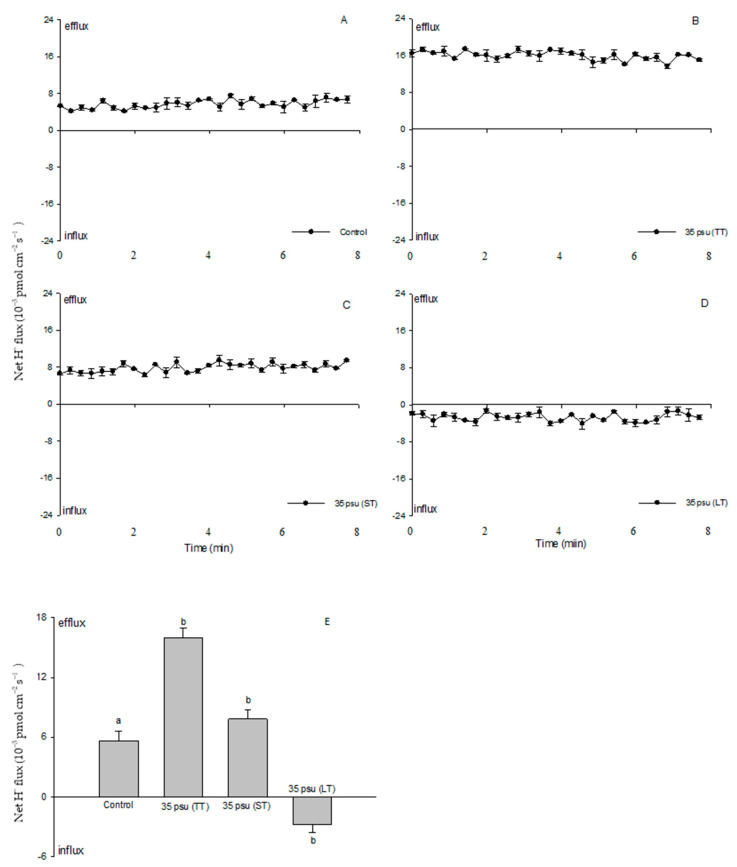
Net fluxes of H^+^ from control (**A**), TT-stressed, (**B**) ST-stressed, (**C**) and LT-stressed (**D**) cells of the estuarine diatom *Coscinodiscus centralis*. A continuous flux recording of 7 to 8 min was conducted for each cell. Each point represents the mean of four to five individual diatoms, and bars represent the standard error of the mean. The mean fluxes of H^+^ within the measuring periods are shown (**E**). Columns labeled with different letters (a and b) are significantly different at *p* < 0.05.

**Figure 4 ijms-24-13683-f004:**
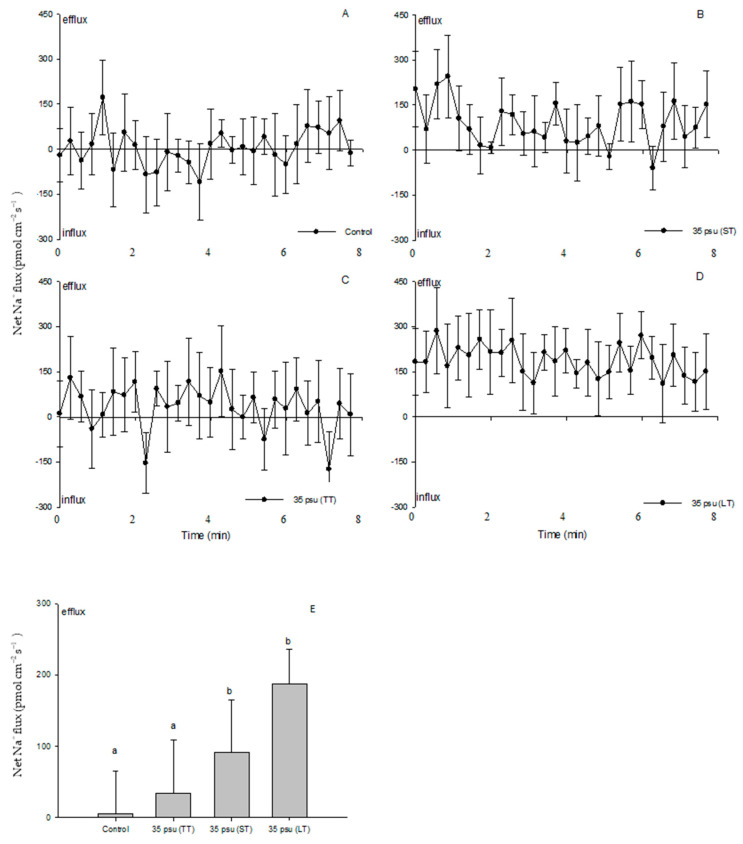
Net fluxes of Na^+^ from control (**A**), TT-stressed, (**B**) ST-stressed, (**C**) and LT-stressed (**D**) cells of the estuarine diatom *Coscinodiscus centralis*. A continuous flux recording of 7 to 8 min was conducted for each cell. Each point represents the mean of four to five individual diatoms, and bars represent the standard error of the mean. The mean fluxes of Na^+^ within the measuring periods are shown (**E**). Columns labeled with different letters (a and b) are significantly different at *p* < 0.05.

**Figure 5 ijms-24-13683-f005:**
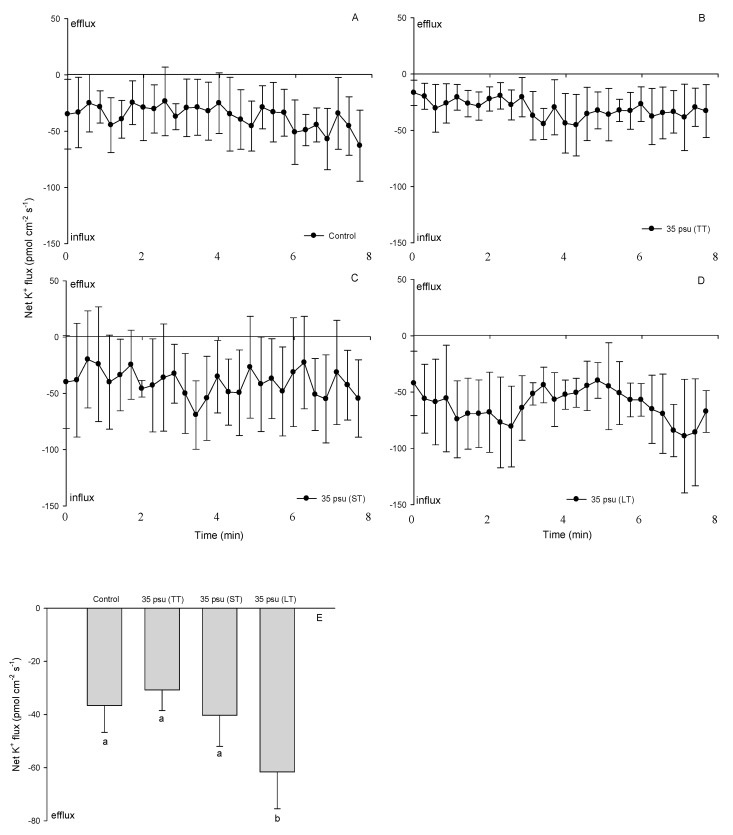
Net fluxes of K^+^ from control (**A**), TT-stressed, (**B**) ST-stressed, (**C**) and LT-stressed (**D**) cells of the estuarine diatom *Coscinodiscus centralis*. A continuous flux recording of 7 to 8 min was conducted for each cell. Each point represents the mean of four to five individual diatoms, and bars represent the standard error of the mean. The mean fluxes of K^+^ within the measuring periods are shown (**E**). Columns labeled with different letters (a and b) are significantly different at *p* < 0.05.

**Figure 6 ijms-24-13683-f006:**
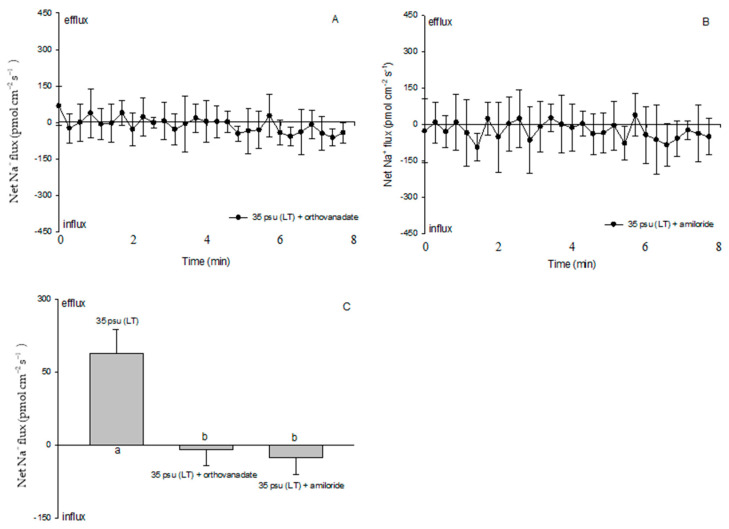
Effects of sodium orthovanadate (100 µM) and amiloride (25 µM) on salt-shock-induced Na^+^ kinetics in the LT-stressed estuarine diatom *Coscinodiscus centralis*. Cells were pretreated with sodium orthovanadate or amiloride for 15 min prior to the flux measurements. (**A**) and (**B**) Na^+^ kinetics recorded in cells pretreated with inhibitors. Each point represents the mean of four or five cells, and bars represent the standard error of the mean; (**C**) shows the mean Na^+^ flux rates in cells pretreated with inhibitors. Columns labeled with different letters (a and b) are significantly different at *p* < 0.05.

**Figure 7 ijms-24-13683-f007:**
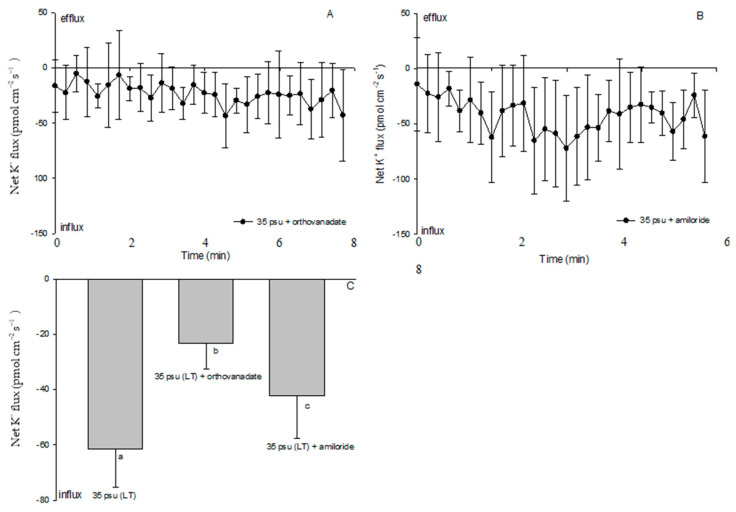
Effects of sodium orthovanadate (100 µM) and amiloride (25 µM) on salt-shock-induced K^+^ kinetics in LT-stressed estuarine diatom *Coscinodiscus centralis*. Cells were pretreated with sodium orthovanadate or amiloride for 15 min prior to fluxes measurements. (**A**) and (**B**) K^+^ kinetics recorded in cells pretreated with inhibitors. Each point represents the mean of four or five cells, and bars represent the standard error of the mean; (**C**) shows the mean K^+^ flux rates in cells pretreated with inhibitors. Columns labeled with different letters (a, b and c) are significantly different at *p* < 0.05.

**Figure 8 ijms-24-13683-f008:**
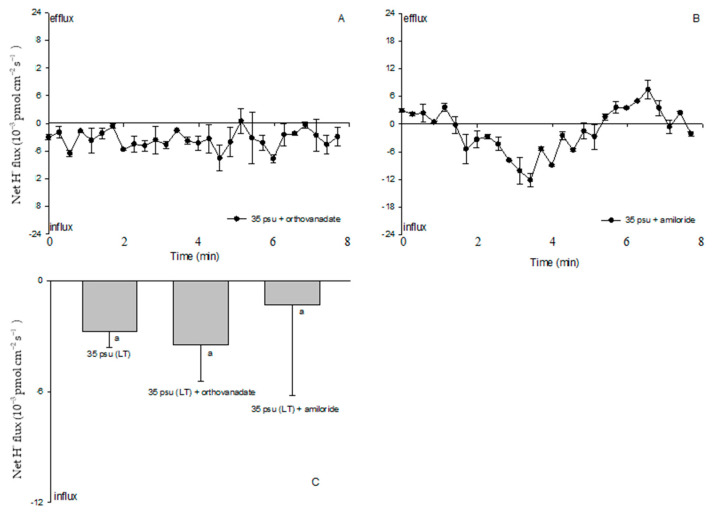
Effects of sodium orthovanadate (100 µM) and amiloride (25 µM) on salt-shock-induced H^+^ kinetics in the estuarine diatom *Coscinodiscus centralis*. Cells were pretreated with sodium orthovanadate or amiloride for 15 min prior to flux measurements. (**A**) and (**B**) H^+^ kinetics recorded after increased salinity with 35 psu, and steady H^+^ fluxes were measured in cells pretreated with inhibitors. Each point represents the mean of four or five cells, and bars represent the standard error of the mean; (**C**) shows the mean H^+^ flux rates in cells pretreated with inhibitors. Columns labeled with the same letter (a) are not significantly different at *p* > 0.05.

**Figure 9 ijms-24-13683-f009:**
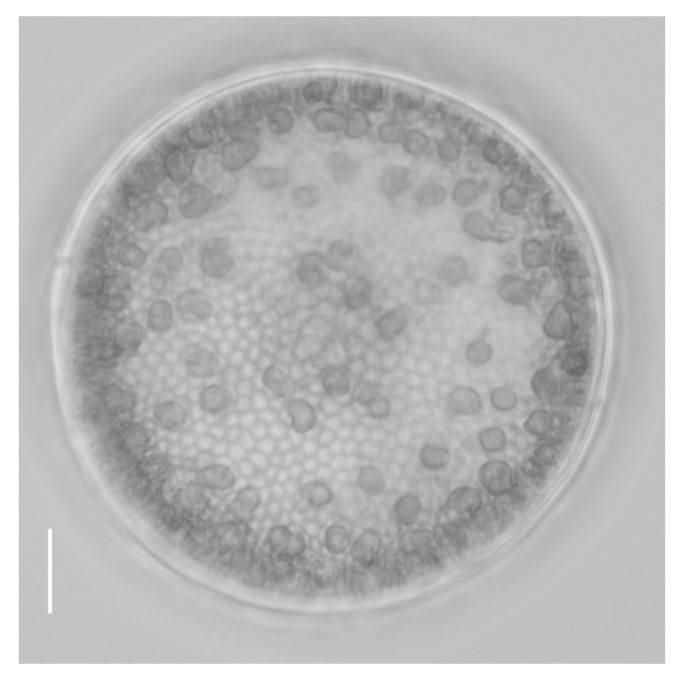
LM micrograph of *C. centralis* (scale bar = 10 μm).

## Data Availability

Not applicable.

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
