# Peer review of "Ion fluxes Involved in the Adaptation of the Estuarine Diatom Coscinodiscus centralis Ehrenberg to Salinity Stress"

_ijms, 2023, doi:10.3390/ijms241813683_

Round 1

Reviewer 1 Report

Paper entitled “Ion fluxes involved in the adaptation of estuarine diatom to salinity stress” presents interesting studies and can be publish after tatin into consideration same correction:

-Introduction should be much more focused on diatoms then like now on green algae.

-Im my opinion Authors should include same additional data about diatom morphology in cultures: valves diameter in each culture type (mean, max, min), frustule picture from culture from LM or SEM microscope.

-Same minor remarks in the ms file.

Author Response

Response to Reviewer 1 Comments

Point 1: -Introduction should be much more focused on diatoms then like now on green algae. The introduction must be improved.

Response 1: To our knowledge, there are few studies on the ionic homeostasis of diatoms. Therefore, we add some diatom results, but also have introduction in green algae, which could be comparable. (Line 58-59) showed as”Levels of intracellular K+ increased significantly with increases in salinity in diatom Chaetoceros muelleri Lemmermann [23].”.

Point 2: -Im my opinion Authors should include same additional data about diatom morphology in cultures:valves diameter in each culture type(meanmaxmin),frustule picture from culture from LM or SEM microscope.

Response 2: We agree and add. (Line 89-92, 274-275) showed as” We also measured the valves diameters of C. centralis in three treatments. The diameters of C. centralis ranged from 52.9 to 71.0 μm, 54.9 to 74.9 μm, and 56.2 to 74.5 μm, with mean diameters of 62.2 ± 5.0 μm, 65.0 ± 5.7 μm, and 64.6 ± 5.5 μm in the TT, ST, and LT treatments, respectively.”, and “Figure 9. LM micrograph of C. centralis (Scale bar = 10 μm)”

Point 3: -Same minor remarks in the ms file.

Response 3: We revised and highlighted these revisions in the revised file.

Reviewer 2 Report

In the reviewed MS an interesting data about the adaptation of estuarine diatom to salinity stress are presented. The obtained results are very important for better understanding the physiology and ecology of marine diatoms. The results supported by figures.  The reference list contains the basic publications on studied topic.

I can recommend the paper for the publication after some corrections.

Major suggestions:

1.      Why did you choose Coscinodiscus centralis for your study? Please, give an explanation in the Introduction.

2.      Please, write the separate final paragraph with the conclusive statements of your investigation.  

3.      Is it possible to include some pictures of Coscinodiscus centralis into the MS?

Minor suggestions:

Lines 14, 44 and further: Write the authors of the taxon at the first mention.

Line 17: Explain abbreviations ST and LT at the first mention.

Figures 3-8: Please, enlarge figures. It is very difficult to examine it.

Lines 265, 313, 314: What symbol did you use before “mol photons m-2 s-1”?

Minor editing  required

Author Response

Response to Reviewer 2 Comments

Point 1: - The introduction must be improved.

Response 1: We improve the introduction, as we response to the reviewer 1. Showed as “To our knowledge, there are few studies on the ionic homeostasis of diatoms. Therefore, we add some diatom results, but also have introduction in green algae, which could be comparable. (Line 58-59) showed as” Levels of intracellular K+ increased significantly with increases in salinity in diatom Chaetoceros muelleri Lemmermann [23].”.

Point 2: -1. Why did you choose Coscinodiscus centralis for your study? Please, give an explanation in the Introduction.

Response 2: Coscinodiscus centralis Ehrenberg is a dominant species in the Jiulong River estuary, and is a good material for studying how estuarine diatoms maintain ionic homeostasis under high salinity conditions.

Point 3: Please, write the separate final paragraph with the conclusive statements of your investigation.

Response 3: We add conclusion, showed as “The alternations of ion fluxes in estuarine diatom with different enhanced salt exposure time were compared. We found that C. centralis cells exhibited marked H+ effluxes after TT and ST treatment. However, a drastic shift of H+ efflux toward an influx was induced in LT treatment. Na+ flux in TT, ST and LT salinity conditions was found to accelerate Na+ efflux. K+ influx showed significant increase in LT salinity condition. Our results indicate that the Na+ extrusion in salt-stressed cells is mainly the result of an active Na+/H+ antiport across the plasma membrane. The pattern of ion fluxes in TT and ST salinity conditions were different from those under LT salinity, suggesting an incomplete regulation of the acclimation process in estuarine diatom under short term salinity stress.”

Point 4: Is it possible to include some pictures of Coscinodiscus centralis into the MS?

Response 4: We add it in Materials and Methods (Line 276-277).

Point 5: Iines 14 44 and further Write the authors of the taxon at the first mention

Response 5: we have revised and highlighted in the revised file.

Point 6: Line 17: Explain abbreviations ST and LT at the first mention.

Response 6: The previous sentence already states, showed as “C. centralis cells exhibited marked H+ effluxes after a transient treatment (TT, 30 min) and short-term treatment (ST, 24 h). However, a drastic shift of H+ efflux toward an influx was induced in the long-term treatment (LT, 10 days).”

Point 7: Figures 3-8: Please,enlarge figures.It is very difficult to examine it.

Response 7: We revised in Figures 3-8.

Point 8: Lines 265, 313,314: What symbol did you use before "mol photons m-2 s-1"?

Response 8: We revised as μ in the revised file.
